# Seasonal Use of Dairies as Overnight Roosts by Common Starlings (*Sturnus vulgaris*)

**Callan Lichtenwalter** [1], **Karen Steensma** [2], **Marcos Marcondes** [1], **Kyle Taylor** [3], **Craig McConnel** [3]
**and Amber Adams Progar** [1,*]

[1] Department of Animal Sciences, Washington State University, Pullman, WA 99164, USA;
   c.lichtenwalter@wsu.edu (C.L.); marcos.marcondes@wsu.edu (M.M.)
[2] Department of Geography and Environment, Trinity Western University, Langley, BC V2Y 1Y1, Canada;
   steensma@twu.ca
[3] College of Veterinary Medicine, Washington State University, Pullman, WA 99164, USA;
   k.taylor@wsu.edu (K.T.)
[*] Correspondence: amber.adams-progar@wsu.edu





**Simple Summary:** Common Starlings are invasive in North America. Starlings congregate in roosts overnight to sleep and will often use structures on dairies as their roost location. While at the roost, starlings defecate on cows, feed, and equipment and eat cattle feed. This behavior can lead to great costs for farmers and potentially the spread of disease. Understanding when starlings use dairies as night roosts can help farmers decide when to implement deterrence methods. To understand their seasonal behavior, starlings were counted around sunrise and sunset on 10 dairies in Whatcom County, Washington and as they flew into barns to roost at sunset at two dairies in the Palouse region of Washington and Idaho. We found that starling counts increased as fall progressed, and that more starlings used dairies as night roosts in winter and spring than in summer and when it was colder or there were fewer hours of daylight. This suggests that starlings prefer to use dairies as night roosts in the late fall, winter, and early spring. With this information, farmers in the United States will know to contact wildlife managers in the summer, so a starling deterrence strategy can be developed before roosts are established in the fall.

**Abstract:** The Common Starling (*Sturnus vulgaris*) is an invasive bird species in North America that can cause damage to dairies. Starlings roost in structures on dairies overnight, defecating on cows, feed, and supplies. To target roosts for effective deterrence, farmers must know what times of the year starling populations are greatest. To test seasonality, two data sets were analyzed. First, birds were counted for 20–30 min at sunrise and sunset on 10 Whatcom County, Washington dairies over four weeks in September and October of 2016. Starling counts were greater in the last week of observations than in weeks one, two, and three. Second, birds were counted at sunset for four weeks during winter 2021 and spring and summer 2022 at two dairies in the Palouse region of Washington State and Idaho. As temperature and minutes of daylight decreased, bird abundance increased. There was also an effect of season, with more birds counted in winter and spring than in summer. These data sets combined suggest a seasonal use of dairies as night roosts by starlings. With this information, farmers in the United States will know to contact wildlife managers in the summer, so a starling deterrence strategy can be developed before roosts are established in the fall.

**Keywords:** deterrence; agriculture; invasive species; bird abundance

## 1. Introduction

With increasing globalization, there has been an explosion in the number of human-introduced invasive species, which has consequences for biodiversity, ecosystem health, human health, and economic interests [1]. In the United States alone, approximately USD

21 billion are spent annually managing biological invasions, with the agriculture sector incurring the largest economic brunt of these invasions [2]. These estimates of the economic costs of invasive species are likely lower than what is actually spent on damages and management due to gaps in knowledge and reporting. In addition to great financial losses within agriculture, invasive species lead to decreases in animal and crop productivity, and thus food security, and can cause stress, disease, and increased energy needs in agricultural animals [3,4].

Although native to Europe, southwest Asia, and northern Africa, Common Starlings (*Sturnus vulgaris*) can be found on every continent except Antarctica [5]. Despite gains on other continents, in their native range, starling abundance has declined, often due to the intensification of farming [6]. Starlings were introduced to the United States in 1890 by immigrants who released the starlings to have a reminder of home [7]. With just 50 breeding pairs released in Central Park at the time, starlings were so successful they reached the west coast of the United States in around half a century [5]. While starling population estimates in North America once reached 200 million, today there are an estimated 86–100 million starlings on the continent, which follows recent declines seen in many other species of passerine birds [5,8,9].

Due to their success, starlings are currently on the IUCN list of the world's 100 worst invasive species with just two other avian species, the Common Myna (*Acridotheres tristis*) and the Red-vented Bulbul (*Pycnonotus cafer*) [10]. Cuthbert and others (2021) wanted to know if the invasive species mentioned on the IUCN list were also the costliest invasive species by comparing them to invasive species in the InvaCost database [11,12]. When using the data from Cuthbert and colleagues to find the greatest estimated costs incurred by invasive animal species adjusted to USD-2017, starlings rank as the 21st costliest invasive animal species and the second costliest avian species behind Rock Pigeons (*Columba livia*). Starlings cause more damage in agriculture than in any other sector, and based on a conservative estimate that starlings cause USD 5 of damage per hectare of agricultural land, they cost American farmers USD 1,138,766,938 annually [11,13].

Although they may provide insect control in a few crops in the United States, starlings are common nuisance birds at dairies and feedlots [14]. Within animal agriculture, starlings can spread disease and eat and spoil livestock feed. Starlings have been found to spread *E. coli* [15–18], *Salmonella* [19–21], and antimicrobial-resistant bacteria [14]. When eating cattle TMR, starlings opt for the high-energy portion of the ration, leaving a more fibrous ration for the cattle [22,23]. Estimates of the costs associated with starlings eating and spoiling feed have been reported as USD 0.92 per cow per day and USD 4.92–6.99 per cwt of milk produced [23,24].

Once starlings have established a roosting site, it is often difficult to remove them. Common Starlings are flexible in choosing a location to establish a roost. Roost site locations are often concentrated in farmland that heavily fragments natural areas and contains a variety of crops as possible food sources, but they can also be found near population centers [25–27] Roosting site fidelity has been reported to be between 48% and 95%, with fidelity increasing with the age of the bird and availability of feed near the roosting site [28–30]. This pattern of roost site fidelity is also seen on a smaller scale, with starlings preferring to stay in one roost site overnight unless their energetic needs increase due to cold weather or a lack of an easily available food source [31]. Of the two other major invasive bird species in the United States, House Sparrows and Rock Pigeons, House Sparrows also exhibit over-nighting roost behavior and have high roost site fidelity, so it can be difficult to dissuade them from a roost site [32]. However, Rock Pigeons do not have high roost site fidelity and will move to a new site when disturbed [33].

Because starlings tend to stay in one area once they have established a roosting site, and an increase in starlings on a farm causes an increase in disease and feed loss [14–24], it is important for farmers to understand when starlings pose the greatest risk on their farms so deterrence measures can be implemented ahead of time. Since starlings are invasive to the United States, they are not considered a protected species under federal

law, so deterrence is acceptable year-round. Although studies have investigated starling movements and implied the increase in starling abundance on dairies in winter in the United States [24], no study has looked at the seasonality of starlings using dairies as roosting sites by taking direct starling counts. To understand their seasonal movements on dairy farms, we analyzed two sets of data collected from dairy farms in Washington State and Idaho. One data set was collected from Whatcom County, Washington in 2016, and the other data set was collected from the Palouse region of Washington and Idaho in 2021 and 2022. Based on anecdotal evidence from farmers, we predicted that the number of starlings roosting on dairies (starling abundance) would be the greatest when the environmental temperature was the coolest and when minutes of daylight were the fewest (late fall, winter, and early spring) and analyzed the two data sets to test this prediction.

## 2. Materials and Methods

### 2.1. Locations and Animals—Whatcom County

Whatcom County is in the northwest corner of Washington State, stretching from the North Cascade region to the lowland delta plane of the Nooksack River. There is an abundance of agricultural land in this region consisting of mostly berry crops and dairy farms. Whatcom County is located in a moist sub-tropical mid-latitude climate according to the Köppen classification system [34]. Farms included in this study were located in either Lynden, Everson, or Ferndale, Washington. Lynden has a population density of 1978 people/km$^2$, Everson has a population density of 1812 people/km$^2$, and Ferndale has a population density of 1345 people/km$^2$ [35].

Dairies located within a mile of fruit crops in Whatcom County, Washington were chosen for this portion of the study. The dairy–fruit crop interface is common in Whatcom County, and both farm types tend to attract pest birds. After interviews with area dairy farmers, ten dairies were selected to determine bird abundance: seven in Lynden; two in Everson; and one in Ferndale. Each dairy had 400–1500 milking cows and housed their cows in freestall barns. Each dairy fed a TMR at least once daily and had open commodity sheds. Care was taken to ensure that the selected dairies were as similar in production practices as possible.

### 2.2. Locations and Animals—Palouse

Located about 633 km southeast of Whatcom county, the Palouse is a region of rolling hills and agricultural land in Southeastern Washington and Northern Idaho. The climate in the Palouse is considered moist continental mid-latitude according to the Köppen classification system [34]. Bird abundance was recorded at the dairies associated with Washington State University in Pullman, Washington, and the University of Idaho in Moscow, Idaho. The two farms are 19.8 km apart. Pullman has a population density of 1853 people/km$^2$, and Moscow has a population density of 2363 people/km$^2$ [35].

The Knott Dairy Center (KDC) is the research and teaching dairy for the Department of Animal Sciences at Washington State University and is located a few miles off campus. At KDC, an average of 180 Holstein lactating cows are housed in four pens with access to freestall barns with compost bedding. Cows were fed a TMR, milked twice a day, and had ad libitum access to water and mineral supplements. The species of birds typically seen roosting in the freestall barns at KDC are Common Starlings and House Sparrows (*Passer domesticus*), both of which are non-native to the United States.

The University of Idaho Dairy Center (UI) is the research and teaching dairy for the Department of Animal, Veterinary, and Food Science at the University of Idaho. An average of 100 lactating cows are housed at UI. The herd consists of Holstein, Jersey, and Holstein × Jersey crossbred cows. The lactating cows are housed in four pens with access to a freestall barn with compost bedding. Cows were fed a TMR daily, milked twice a day, and had ad libitum access to water and mineral supplements. The same species of birds seen roosting at KDC also roost in the freestall barn at UI.

## 2.3. Bird Counting Methods—Whatcom County

Farmers in Whatcom County observed an increase in pest birds on their farms in the fall as temperatures began to drop, with a peak of bird activity occurring during the coldest times of the year. To quantify the abundance of pest birds on their farms, researchers performed point counts of birds on the ten dairies during September and October of 2016. An observer visited each farm twice a day, once per week, for four consecutive weeks. The trained observer remained in a vehicle while determining the bird abundance in order to provide a blind. Cars were positioned about 1 m from the edge of the barn being observed so that the observer could view both inside the barn, the feed bunk and rafters, and the roof outside of the barn with just the unaided eye. Bird abundance was measured for 20–30 min within two hours of sunrise and sunset on each observation day to observe starlings coming and going from roost sites. These observation times were chosen because starlings had been observed coming and going from roosts within barn structures almost exclusively at sunrise and sunset. The observer monitored feed bunks and farm building structures for bird activity, and if more than two dozen birds were counted roosting in a building, the total number of birds occupying the building was extrapolated based on the known area of the roost. Common Starlings were the focus of the observations, but measures of abundance of other passerine birds such as Rock Pigeons and Eurasian Doves (*Streptopelia decaocto*) were also recorded. In a few cases, the bird abundance for one location was not measured during the pre-determined times, or it was conducted on separate days during the week. In these cases, the data were removed from our analysis.

## 2.4. Bird Counting Methods—Palouse

To estimate bird abundance, the number of birds entering the freestall barns to roost overnight was recorded. Data collection periods occurred during winter, spring, and summer. The winter collection period was from mid-November to mid-December of 2021, the spring collection period was during March of 2022, and the summer collection period was during July of 2022. Each collection period occurred over four consecutive weeks, and observations were recorded one night a week at both university farms.

Approximately 45 min before sunset, a trained observer counted the number of birds already inside the freestall barns and then stood outside the freestall barns to count birds as they flew in to roost for the night. This was the time starlings had been observed entering the freestall barns to roost overnight on multiple previous occasions. The observer stood approximately 4 m away from the freestall barn being observed and counted birds with the unaided eye. When large groups of birds flew into the freestall barns, the observer would count a portion of the group and extrapolate the total number of birds. For example, as birds were flying in, the first 10 could be counted. Based on the proportion of the group those 10 birds occupied, for example, 10%, a total abundance could be estimated at 100 birds. Observations continued until approximately 20 min after sunset, when it was too dark to observe and make accurate estimates.

## 2.5. Weather Data

Data on daily high temperature and minutes of daylight were retrieved from a weather website (https://www.weatherunderground.com; accessed on 15 August 2022). The weather data were collected from the Abbotsford International Airport station in Abbotsford, British Columbia, Canada for the farms located in Lynden and Everson, Whatcom County, Washington. Lynden is 19.2 km and Everson is 24.2 km from this weather station. Weather data for the Ferndale, Whatcom County, Washington farms were collected from the Bellingham International Airport in Bellingham, Washington. Ferndale is 11.6 km from this weather station. For the Palouse locations, the weather data were collected from the Pullman–Moscow Regional Airport, which is 9.8 km from the UI dairy and 13.8 km from KDC. Multiple observations occurred per week during the Whatcom County study, so the average temperature for this location was represented by a weekly average of the daily high temperature from both weather stations used for the region. Since one day of observation

occurred per week for each of the two Palouse locations, the daily high temperature on the day of observation at each location was used to represent weekly temperature for this portion of the study.

*2.6. Statistics*

The data for Whatcom County and the Palouse were analyzed separately. For the Whatcom County data, we used a PROC GENMOD procedure of SAS (Version 9.4; Cary, NC, USA) with a Poisson distribution and farm ID as a repeated measure within week. Bird abundance and starling abundance were analyzed as separate, dependent variables. Bird abundance included starlings and all other passerine birds counted during the observation period. Starling abundance was only the number of starlings counted during the observation period. Differences in LS means were used to assess differences in abundance each week.

In the Palouse data set, one outlier in bird abundance was identified by graphing studentized residuals. Since the number of observations in this data set was small (n = 24), statistics were run with and without the outlier. Bird abundance was analyzed using PROC GLIMMIX, run separately with temperature, season (winter, spring, summer), or minutes of daylight as the independent factor. Farm and week were added as random effects. Differences in LS means were used to assess differences in abundance in relation to different temperatures, minutes of daylight, and season.

## 3. Results

Starlings were the most frequently observed bird each week at the Whatcom County dairies, although their presence ranged from 0 to 100 percent of observed birds (Table 1). Average weekly temperature in Whatcom County decreased with each successive week (Figure 1, Table 1).

**Table 1.** Total bird and starling abundance on Whatcom County dairies.

| Week | Total Starlings/Total Birds | Range: Total Starlings/Total Birds | Average Weekly Temperature | StDev *. of Weekly Temperature |
|------|------|------|------|------|
| 1 | 0.71 | 0.18–1.0 | 20.4 | 2.01 |
| 2 | 0.80 | 0.05–1.0 | 18.6 | 3.61 |
| 3 | 0.61 | 0.00–1.0 | 16.8 | 1.24 |
| 4 | 0.90 | 0.55–1.0 | 14.2 | 1.37 |

Proportion of all birds observed at Whatcom County dairies each week of the study that were starlings and the average weekly temperature (°C). Week 1 was 17 September–23 September 2016. Week 2 was 24 September–1 October 2016. Week 3 was 2 October–8 October 2016. Week 4 was 9 October–16 October 2016, * StDev. = Standard Deviation.

In Whatcom County, week had an effect on total bird abundance. The bird abundance for week four was greater than that of week one ($p = 0.01$, n = 37, df = 3), week two ($p < 0.0001$, n = 37, df = 3), and week three ($p = 0.02$, n = 37, df = 3; Figure 1). No differences were detected in bird abundance among weeks one, two, and three. Week also had an impact on starling abundance. Starling abundance showed the same pattern as total bird abundance, with week four having a greater number of starlings compared to week one ($p = 0.004$, n = 37, df = 3), week two ($p < 0.001$, n = 37, df = 3), and week three ($p = 0.01$, n = 37, df = 3; Figure 1). No difference was detected in starling abundance among weeks one, two, or three.

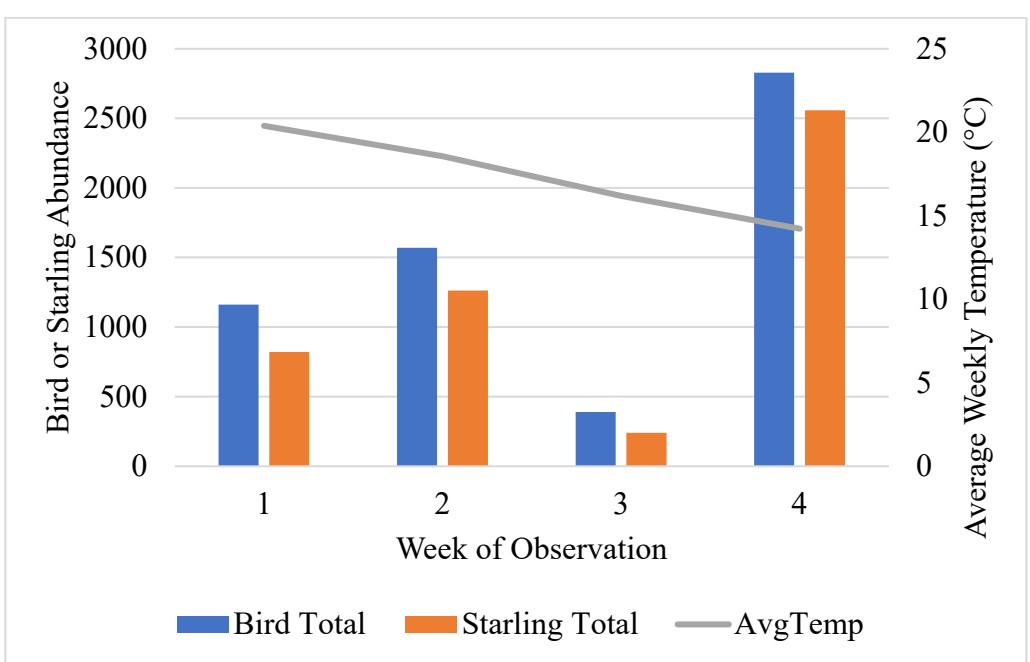

**Figure 1.** Changes in total bird and starling abundance and average weekly temperatures on Whatcom County dairies from mid-September through mid-October. Bird and starling abundance in Whatcom County, Washington from mid-September through mid-October 2016. Temperature is represented through average weekly temperature (°C). Week 1 was 17 September–23 September 2016. Week 2 was 24 September–1 October 2016. Week 3 was 2 October–8 October 2016. Week 4 was 9 October–16 October 2016.

To visualize seasonal changes, bird abundance, temperature, and minutes of daylight were averaged across winter, spring, and summer (Table 2). Excluding the outlier from the data set did not affect the results. With the outlier included in the data set, temperature ($p$ = 0.0002, n = 24, df = 1), season ($p$ = 0.004, n = 24, df = 2), and minutes of daylight ($p$ = 0.008, n = 24, df = 1) all had an effect on bird abundance. As temperature and minutes of daylight increased, bird abundance decreased, and more birds were counted in winter and spring than in summer (Figures 2 and 3). Without the outlier, temperature ($p$ < 0.0001, n = 23, df = 1), season ($p$ = 0.002, n = 23, df = 2), and minutes of daylight ($p$ = 0.001, n = 23, df = 1), each still had an effect on bird abundance.

**Table 2.** Average starling abundance, temperatures, and minutes of daylight in winter, spring, and summer at the KDC and UI dairies.

| UI | Avg [1]. Birds | StDev [2]. Birds | Avg. Temp [3] | StDev. Temp | Avg. MOD [4] | StDev. MOD |
|---|---|---|---|---|---|---|
| Winter | 818 | 274.7 | 6.9 | 5.4 | 529.8 | 13.1 |
| Spring | 839 | 656.4 | 11.8 | 5.9 | 716.3 | 30.6 |
| Summer | 6 | 3.2 | 26.5 | 3.9 | 935.3 | 12.7 |
| KDC | | | | | | |
| Winter | 523 | 181.8 | 7.1 | 6.4 | 533.8 | 16.2 |
| Spring | 716 | 439.2 | 8.2 | 4.2 | 730 | 30.2 |
| Summer | 74 | 45.6 | 27.6 | 3.1 | 932.8 | 11.6 |

Average number of birds, temperature (°C), and minutes of daylight at each Palouse dairy during winter, spring, and summer. KDC is Knott Dairy Center at Washington State University in Pullman, Washington, USA. UI is the University of Idaho Dairy Center in Moscow, Idaho, USA. [1] Avg. = Average, [2] StDev. = Standard Deviation, [3] Temp = Temperature, [4] MOD = Minutes of Daylight.

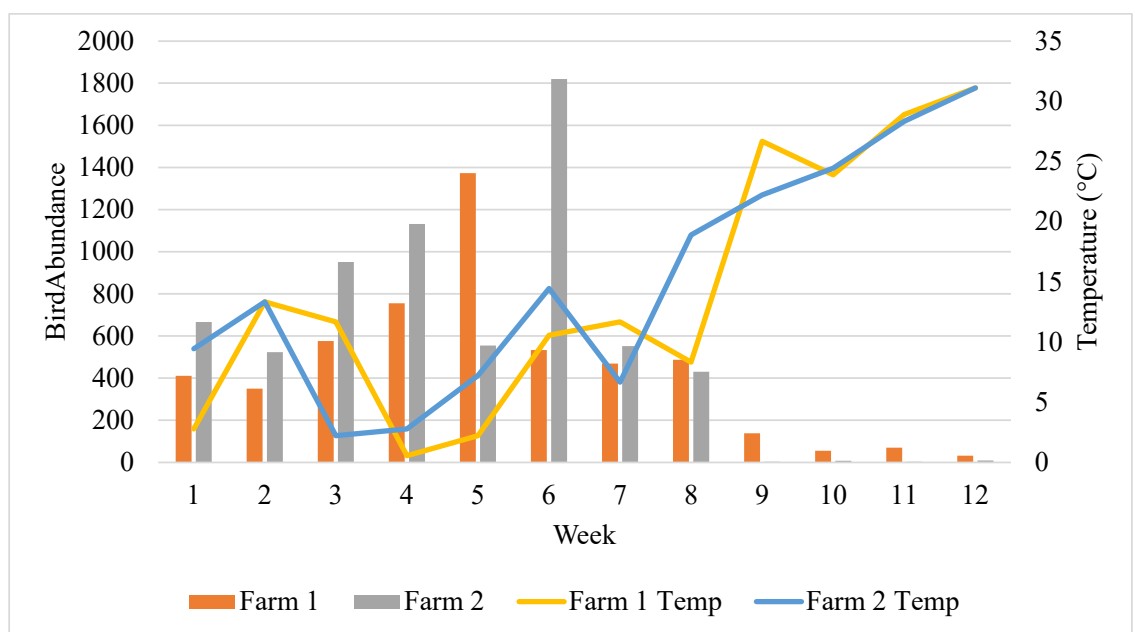

**Figure 2.** Starling abundance and temperatures at the time of bird abundance measurements at the KDC and UI dairies. Bird abundance from Knott Dairy Center at Washington State University in Pullman, Washington, USA (farm 1) and The University of Idaho Dairy Center in Moscow, Idaho, USA (farm 2), and the respective temperature (°C) on the day of bird abundance measurements, across 12 weeks (four consecutive weeks each in winter 2021 (weeks 1–4, mid-November–mid-December), spring 2022 (weeks 5–8, March), and summer 2022 (weeks 9–12, July)).

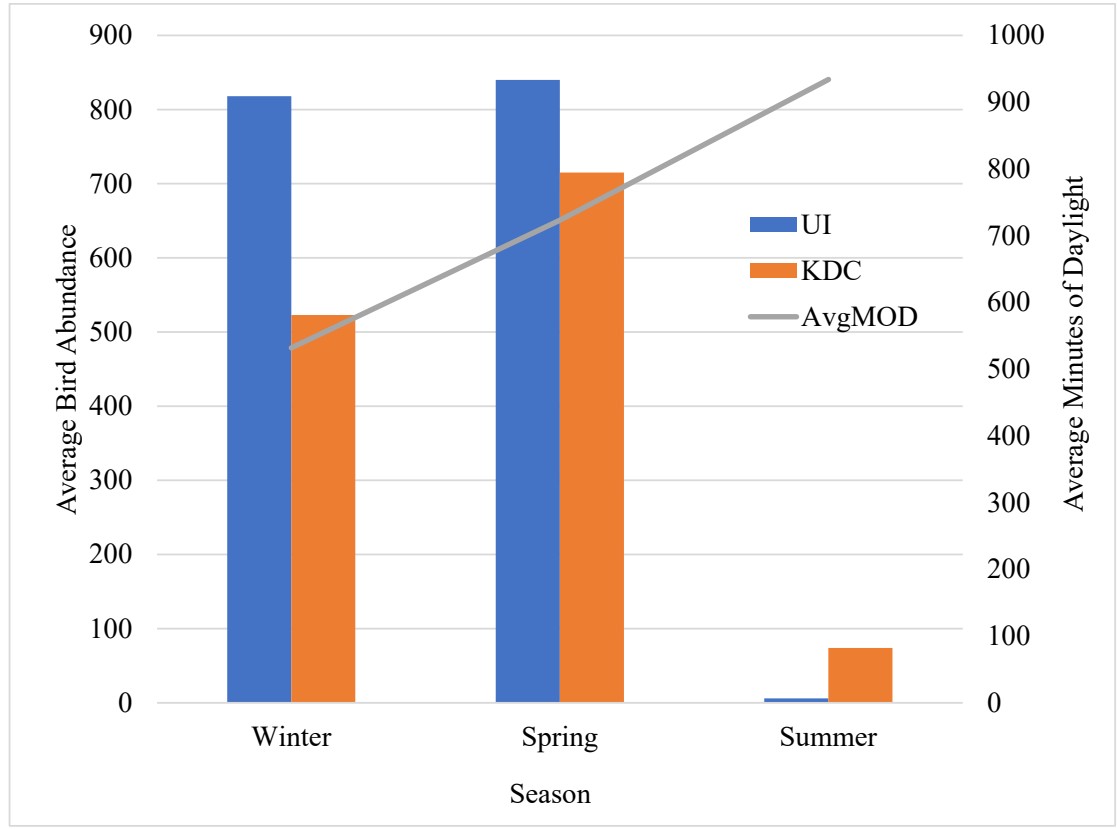

**Figure 3.** Average starling abundance and minutes of daylight in winter, spring, and summer at the UI and KDC dairies. Average bird abundance and minutes of daylight (AvgMOD) at the University

of Idaho Dairy Center in Moscow, Idaho, USA and the Knott Dairy Center at Washington State University in Pullman, Washington, USA in winter (mid-November–mid-December 2021), spring (March 2022), and summer (July 2022). Each seasonal bird abundance average was calculated from four bird abundance measurements at each location. The seasonal minutes of daylight averages were calculated from the minutes of daylight on the eight days (four days at each location) of bird abundance measurements each season.

## 4. Discussion

Starlings were the most abundant bird species observed at the Whatcom County dairies each week. Despite the presence of other bird species that are known to be agricultural pests [36–39], starlings represented between 61% (week 3) and 90% (week 4) of all birds counted. The dominance of starlings as the most common avian pest species on Whatcom County dairies is consistent with previous data from surveys of Washington State dairy farmers [24,40].

In the fourth week of the study in Whatcom County, more birds and starlings were observed than in weeks one through three. The observed increase in birds and starlings could be due to a decrease in temperature over the four weeks of data collection, although the data did not allow us to confirm this hypothesis. Bird abundance did not follow an observable pattern over the four weeks of observation. An explanation is that the transient nature of several migratory bird species in fall can lead to great variation of on-farm bird abundance as birds stopover at a location enroute to their final winter migration destination [41]. Starlings above 40° longitude begin to migrate in early fall [42], and since all of Washington State is north of this latitude, it is understandable that starling abundance would begin increasing over the study period (mid-September through mid-October) but continue to fluctuate.

On both Palouse dairies, starlings were the most common roosting bird. On each observation day, starlings were seen forming murmurations (large flocks of starlings flying together synchronously) close to sunset. The size of starling murmurations has been shown to increase from October through February [43]. After several passes over the dairy, small groups would break off and fly into the freestall barns to roost in the rafters and other open spaces. Starlings are cavity nesters, so when natural roosts become more exposed as vegetation decreases in winter, open air structures such as freestall barns become ideal nesting sites [5]. Starlings are also known to be aggressive and have been seen displacing other birds for preferred nesting sites [5]. This is likely why starlings are the predominant roosting species on dairies despite the variety of species seen during the day.

In the data set from the Palouse dairies, a relationship between bird abundance and temperature was detected; when temperature decreased, bird abundance increased. Although the spring observation period had slightly greater average temperatures and bird abundance (10 °C, 777 birds) than the winter observation period (7 °C, 671 birds), the spring had several wet, snowy days that might have increased the starlings' motivation to roost in the freestall barns. Poor weather conditions have been known to temporarily alter the behavior of birds so that they can maintain more optimal nesting conditions [44]. Although temperature offered a possible explanation for changes in bird abundance in this study, future studies could use a more robust measure of weather to better understand all of the climactic factors that motivate starlings to roost on dairies. Minutes of daylight also had an effect on bird abundance. It is unclear exactly how daylength affects the number of starlings roosting on dairies, but there is evidence that decreasing daylength can alter roost arrival and departure times relative to sunrise and sunset in other species of birds [45,46].

Season had an effect on starling abundance, and season encompasses both temperature and minutes of daylight, among other factors, into one measure. Migration behavior can be highly variable between years and individuals, but the starling fall migration typically begins in September and continues into early December, and their spring migration begins between mid-February and late March, [42,47,48]. This is consistent with our findings that there were more starlings counted on the Palouse dairies in winter and spring than in

summer, so Washington State and northern Idaho could be winter migration destinations for starlings. Diet may also play a role. Starlings are omnivorous and switch from an insectivorous diet in the summer to an herbivorous diet in the winter [49]. However, changes in foraging behavior do not fully explain night roosting behavior on dairies because starlings tend to do most of their foraging during the day [50], and shelter also likely plays a key role. In the winter when vegetation dies away, the natural cavities that starlings roost in become more exposed to the elements, driving them to find alternative options for shelter. Freestall barns on dairies provide a protected cavity for starlings to roost in that allows them to escape precipitation and wind. As a source of both shelter and food, dairies are an ideal location for starlings to night roost during times of the year when shelter and food are not readily available in the natural environment.

Accurately estimating starling abundance on dairy farms can be difficult. Birds are highly mobile, and individuals look indiscernible to the human eye. Large-scale measures of abundance of wild starlings have been conducted using cameras [51] and point counts [52]. Research estimating starling abundance on dairies has been limited, but some of the methods include taking multiple abundance measurements at high-use areas over one day in summer and one day in fall and choosing the highest abundance for each day [53] or through farmer estimates [24,40]. The former method only offers a snapshot of starling abundance over two days of the year and does not consider the variability in starling movement over the year [41]. The latter method can be affected by human error, and when the data are reported as ranges of starlings, a degree of accuracy is lost. The methods described in this study are more direct counts of starlings, and a way to estimate starlings using dairy structures as roosting sites. Both methods have the potential for a slightly inflated abundance of starlings, since birds can come and go as they please. That is why establishing a time limit for the methods was important. In the Palouse locations, stopping measurements of abundance 20 min after sunset reduced the likelihood that starlings would be recounted if they left the freestall barn to feed and then came back to roost. This is not a perfect method, and it has not been validated, but it is a simple, scientific method that allows us to estimate the number of roosting starlings on a dairy.

Together, these two studies add further evidence of starling attraction to dairies. Both studies documented starlings as a dominant avian pest species on dairies, either in general (Whatcom County) or specifically for overnight roosting (Palouse). With open structures and abundant food sources, dairies make excellent locations for starlings to spend winters when food and roosting site availability decreases. The dairies in Whatcom County saw an increase in starlings as fall progressed, and the Palouse dairies saw an increase in starlings roosting in winter and spring and as temperature and minutes of daylight decreased. Together these datasets demonstrate the seasonality of starling movement on dairies. Further research should be conducted to examine if starlings use pasture-based, freestall, tie stall, and open-lot dairy systems as roosting sites differently. Studies determining starling abundance at dairies should also be conducted at larger, commercial dairies, over multiple years, and in different regions of North America to examine how these differences impact starling abundance. Although one ideal starling deterrence method has not yet been established, understanding when starlings use dairies as night roosts allows farmers to contact local wildlife managers early enough so that a deterrence plan can be developed before a roost is established for the season.

## 5. Conclusions

Starlings were the most common species of bird found on dairies in both Whatcom County, Washington and the Palouse region of Washington and Idaho. Bird abundance increased on Whatcom County dairies as fall progressed. Temperature, minutes of daylight, and season all had an effect on starling abundance on dairies on the Palouse. Together, these data suggest that starlings are more attracted to using dairies in the United States as a night roost during late fall, winter, and early spring, likely because dairies are a source of abundant food and shelter. This information helps farmers decide the best time to

implement starling deterrence strategies so that they can be most effective in keeping starlings from establishing night roosts on their farms.

**Author Contributions:** Conceptualization: K.S. and A.A.P.; Methodology: C.L., A.A.P. and K.S.; Investigation; C.L. and K.S.; Formal Analysis: C.L., A.A.P. and M.M.; Writing—Original Draft Preparation: C.L.; Writing—Review and Editing: C.L., A.A.P., K.S., M.M., K.T. and C.M.; Supervision: A.A.P., M.M., K.T. and C.M. All authors have read and agreed to the published version of the manuscript.

**Funding:** No sources of funding to acknowledge for this research.

**Institutional Review Board Statement:** All research was approved by the Washington State University Institutional Animal Care and Use Committee protocol number 6969.

**Informed Consent Statement:** Not applicable.

**Data Availability Statement:** All data are available on request from the corresponding author.

**Acknowledgments:** We would like to acknowledge all of the students that helped collect bird abundance data. Thank you to the owners of the dairies in Whatcom County and to the staff at the Washington State University and University of Idaho dairies for their cooperation.

**Conflicts of Interest:** The authors declare no conflict of interest.

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
