# Peer review of "Seasonal Use of Dairies as Overnight Roosts by Common Starlings (Sturnus vulgaris)"

_2673-6004, doi:10.3390/birds4020018_

Round 1

Reviewer 1 Report

The study has the clear objective of understanding the seasonal movements of the Starling on dairy farms analysing two sets of data collected in Washington State and Idaho. The research activity could be better contextualized by including a comparison with studies dealing with the assessment of damage to farms in relation to landscape and territorial factors and their availability of food. Here some suggestions:

Bozzo F, Tarricone S, Petrontino A, Cagnetta P, Maringelli G, La Gioia G, Fucilli V, Ragni M. Quantification of the Starling Population, Estimation and Mapping of the Damage to Olive Crops in the Apulia Region. Animals. 2021; 11(4):1119. https://doi.org/10.3390/ani11041119

P. Clergeau, D. Fourcy, Effects of landscape homogeneity on starling roost distribution, Agriculture, Ecosystems & Environment, 2005; Volume 110, Issues 3–4. https://doi.org/10.1016/j.agee.2005.04.022.

Author Response

Responses to reviewers are organized by reviewer.  Summaries of reviewer suggestions are bolded, while author responses are italicized.

Reviewer One

Research could be contextualized by including studies dealing with assessment of landscape and territorial factors and food availability.

Thank you for the suggestions.  We added two sentences in lines 76-79 that discuss how differences in landscape factors can influence starling attraction to a certain area as a preferred roost site. The studies you listed were interesting, but not our area of expertise, so please advise if further work needs to be done to address your concerns.

Bozzo, F., S. Tarricone, A. Petrontino, P. Cagnetta, G. Maringelli, G. La Gioia, V. Fucilli, and M. Ragni. 2021. Quantification of the starling population, estimation, and mapping of the damage to olive crops in the Apulia region. Animals. 11(4):1119.

Clergeau, P. and F. Quenot. 2007. Roost selection flexibility of European starlings aids invasion of urban landscape. Landscape Urban Planning. 80(1-2):56-62.

Clergeau, P. and D. Fourcy. 2005. Effects of landscape homogeneity on starling roost distribution. Agric. Ecosyst. Environ. 110(3-4):300-306.

Reviewer 2 Report

This is a well-written manuscript of a short and concise study performed over a specific time period in a specific geographical location. The information presented is novel to the particular region and can guide local stakeholders in making decisions for the area. 

Introduction:

The introductions summarizes the issue at hand well and sets the stage by describing the species involved, the context and the research questions.

Methods:

The methods are well-described and can be replicated as needed.

Results:

The results are presented in an easy to understand manner.

Discussion:

The discussion summarize the findings and relate them back to the research objectives and questions. However, more direction could be provided for future studies, as there is scope for extension of this study across multiple seasons to assess other factors that may affect starling roosting habits, or replicated in similar or distinctly different geographic locations, e.g. intensive vs extensive dairy systems/farms. 

Author Response

Reviewer Two

Provide more direction for future studies.

Thank you for your suggestions. Additional suggestions for further research were added to the last paragraph of the discussion section.

Reviewer 3 Report

Thanks for giving me opportunity to review the manuscript. Overall manuscript is good and have high scientific merit. I have following observations.

Introduction: Overall length is good but required to frame hypothesis and research questions. Include the time biases in the estimate.

Methods; Profoundly sound but need to define the point count method for details read the paper. Tiwari G, Pandey P, Kaul R, Lee H, Singh R (2022). Time-of-day bias in diurnal raptors in arid region of Rajasthan. Acta Ecologica Sinica (In press). https://doi.org/10.1016/j.chnaes.2022.07.005

Results; Need to revise with graphical representation cut the text in short.

Discussion is good quality

Author Response

Reviewer Three

Intro: Include time biases in the estimate.

The authors were uncertain how to address this suggestion, particularly in reference to the “estimate”. Upon looking up time biases, it does not appear that time biases apply to this paper.  We chose the time to perform our counts because sunset was always the time when the starlings came to roost in the freestall barns. Please provide further detail if this does not address your concern.

Methods: Define the point count method.

Added additional details in both paragraphs describing how abundance was determined. If further detail is required, please advise.

Results: Need to revise with graphical representation cut the text in short.

The authors were uncertain what revisions are needed to address this comment. Please advise.
